# Young Women’s Needs Regarding Sexual Preventive Behaviours and Unwanted Pregnancies

**DOI:** 10.3390/healthcare12040425

**Published:** 2024-02-07

**Authors:** Ruben Martin-Payo, María del Mar Fernandez-Alvarez, Xana Gonzalez-Mendez, Aránzazu Muñoz-Mancisidor, Olga Lopez-Dicastillo

**Affiliations:** 1Facultad de Medicina y Ciencias de la Salud, Universidad de Oviedo, 33006 Oviedo, Spain; martinruben@uniovi.es (R.M.-P.); gonzalezxana@uniovi.es (X.G.-M.); munozaranzazu@uniovi.es (A.M.-M.); 2PRECAM Research Group, Instituto de Investigación Sanitaria del Principado de Asturias, 33011 Oviedo, Spain; 3Atención Primaria, Área Sanitaria 3, Servicio de Salud del Principado de Asturias, 33401 Aviles, Spain; 4Unidad Materno-Infantil, Área Sanitaria 3, Servicio de Salud del Principado de Asturias, 33401 Aviles, Spain; 5Departamento de Ciencias de la Salud, Facultad de Ciencias de la Salud, Universidad Pública de Navarra—UPNA, 31008 Pamplona, Spain; olga.lopezdicastillo@unavarra.es; 6IdiSNA—Instituto de Investigación Sanitaria de Navarra, 31008 Pamplona, Spain; 7CreaP Research Group, Universidad Pública de Navarra—UPNA, 31008 Pamplona, Spain

**Keywords:** pregnancy, unwanted, women, health promotion

## Abstract

Unwanted pregnancies are considered a public health problem that affects women’s mental health and quality of life. The aim of this paper was to access university students’ understanding and behaviours regarding unwanted pregnancies and identify their needs to prevent them. Qualitative descriptive design was used, and 13 semi-structured interviews were carried out. Women between 18 and 20 years old participated. They discussed a lack of training for themselves, their partners, and their families, their desire to have access to non-in-person health care resources, and their belief that contraception was expensive. Emotional aspects were relevant, affecting the way communication is established with those close to them and with health professionals. Despite the existence of access to sexual health resources, the findings show the existence of needs related to the prevention of unwanted pregnancies. The findings are presented grouping the main identified needs as “related to capability”, “related to opportunity”, and “related to motivation”. Among the aspects to consider when designing interventions to prevent unwanted pregnancies are the feelings shown by these women, the inclusion of couples and family members in educational programs, and access to non-face-to-face health resources and less expensive contraceptive methods. Interventions for social support and understanding of women are necessary both to prevent unwanted pregnancies and to support adolescents with unwanted pregnancies by avoiding criminalization or blame.

## 1. Introduction

Pregnancies during adolescence constitute a major public health problem today. Approximately 16 million women aged 15–19 years give birth each year worldwide, mostly in low- and middle-income countries [1] due to the fact that people in higher-income countries have better access to sexual and reproductive health care [2]. Despite this, it is not exclusive to developing countries. In Spain, according to the National Institute of Statistics, 7.83% of women under 24 years give birth [3].

In Spain, the Organic Law on Sexual and Reproductive Health and the Voluntary Interruption of Pregnancy 2/2010, of 3 March [4], includes the obligation of the public authorities to develop health, educational, and social policies that guarantee health information on contraception and safe sex, to prevent UP and sexually transmitted infections. In addition, each region can organize its own specific plans. For example, in Asturias, sexual health has been configured as one of the priority lines of the Health Plan for the period 2019–2030 [5].

Unwanted pregnancies (UPs) generally have a negative impact on young women’s mental health, causing psychological distress [6] and affecting their quality of life [7]. They are most frequently observed in the age range of 18–24 years [8]. While a declining trend has been observed, as aforementioned, UP can still be considered a public health issue that needs to be addressed [9].

Previous studies examining behaviours related to UPs, or sexually transmitted infections, include factors that have been associated with a higher risk of UP [10,11,12]. For example, Liuccio et al. [10] conclude that a low-risk perception was significantly associated with frequency of condom use. In a similar sense, Stoklosa et al. [11] conclude a risky sexual behaviour tendency in university students was related with increased probability of UP, for example, participating in sexual intercourse with the lack of contraception [11]. Also, the combination of participation in sexual intercourse and alcohol consumption is related with increased risk of UP [11,12]. However, these studies have had limited success as they lack a comprehensive view of the young women’s own views. 

The Behaviour Change Wheel theory (BCW) is a theoretical framework [13] that can help to better understand and change behaviours that end up in an UP. The central components of this framework are capability, opportunity, motivation, and behaviour (COM-B). The BCW provide a comprehensive framework for assessing population needs. Michie et al. [13], authors of the model, indicate that a change in behaviour requires changes in at least one of the three central components and determines which interventions may be implemented. Therefore, identifying what people need to change is essential to achieve the desired change in behaviour. 

This model has proven useful to understand women’s choices and access to contraception [14] or to design interventions to improve sexual health service use [15]. Likewise qualitative research can provide access to information that quantitative methods cannot. Qualitative methodologies help to understand how individuals make choices regarding their health, considering the contexts they live in [16]. Therefore, qualitative approaches seem appropriate for understanding people’s daily life experiences, specifically behaviours related to UPs during adolescence [17].

Given that no studies have been found that evaluate the needs of adolescents to prevent UPs in Spain, we plan this research to answer the following question: What are the needs of adolescent-aged university students to prevent unwanted pregnancies? The hypothesis that this study tried to contrast is that, even despite the existence of health resources, there are still uncovered needs in the university population in relation to UPs. The aim was to access university students’ understanding and behaviours regarding unwanted pregnancies and identify their needs to prevent them.

Using the BCW theory together with a qualitative approach, this study set out to access university female students’ understanding and behaviours regarding UPs, and ultimately to identify their needs to prevent them.

## 2. Materials and Methods

A qualitative descriptive design was used to gain a comprehensive understanding of the phenomenon of UP among female university students. The qualitative descriptive design is suitable for studies that need to provide descriptions of experiences and perceptions in the area under investigation but do not require a deeply theoretical context [18]. This type of design permits access to participants views as they present them, recognizing the subjective nature of the questions under research [19]. This is particularly relevant in healthcare research, which is commonly concerned with how participants experience health and illness and the associated healthcare interventions [18]. This approach is recommended when it is necessary to access the subjective nature of the problem, with a primary focus on enhancing practice or obtaining data for use in mixed methods studies to design and evaluate interventions [18]. Therefore, this type of design permits not only access to participants’ views but to data to enhance sexual health promotion practices and develop interventions as well.

### 2.1. Sample

In qualitative designs, no formal sample size calculation is conducted. Data collection ends when data saturation is reached, with no more than 60 participants expected [20]. Therefore, participants were selected using purposeful sampling [21]. It combined snowball recruitment methods with face-to-face requests. Participant recruitment was undertaken between November–December 2021. 

The following inclusion criteria were considered: (i) First-year female university students of a Spanish university; (ii) up to 24 years old [22]; (iii) not previously or currently pregnant. 

A sample of 13 female students were interviewed during January 2022 (see participants’ ages and studies they were undertaking in Table 1). They were provided with written and verbal information about the study and those who agreed to participate signed a consent form. The selection of participants finished when discourse saturation was reached [23].

### 2.2. Data Collection 

Semi-structured interviews [24] were carried out using a conversational style. The mean time of the interviews was 39.3 min (SD = 1.74) Specifically, an initial prompt was presented, and the answers provided were further explored using a topic guide when the participants’ answers were short or vague (Table 2). The questions included aspects identified in the literature to be relevant in addressing UPs. The list of potential questions was reviewed by the research team members to ensure clarity, relevance, and completeness. This list was used as the preliminary guide, and the possibility of reformulating the questions as the interviews took place was considered by the research team. Participants responded without difficulties and no changes were necessary. The interviews were digitally recorded and transcribed verbatim.

Due the sensitive nature of the topic addressed, and in order to empathize with the participants and to avoid answers that could be conditioned by feelings such as embarrassment, the interviews were conducted by a 32-year-old woman. The interviewer was a nurse specialized in family and community care, with a high level of experience in the subject, not only from a scientific perspective but also from a real-life perspective as she works in a paediatric and adolescent primary care unit. The interviews took place face-to-face or by video call, depending on the preferences of the participants.

### 2.3. Ethical Considerations

Ethical approval was obtained from the Ethical Committee for Research of the Principado de Asturias (Ref. 2021.263). Participants received detailed information about the study before obtaining consent. All students were informed that participation in the study was completely voluntary.

### 2.4. Data Analysis

A framework analysis approach was used for data analysis following the phases of familiarization with the data and the theoretical framework, the codification of the transcribed interviews, and the interpretation and grouping of the codes into categories [25]. To familiarize themselves with the data, every interview was read at least 3 times. The first reading helped to identify words that were related to the understanding of UPs and the participants’ prevention behaviours. In a second reading, codes were created to describe the content of each fragment of the interviews. The codes were afterwards grouped into categories and themes according to the concepts of the COM-B model [13] with flexibility to identify possible new categories, if the data required it, although no new categories were needed. Two researchers analysed the data separately, and disagreements were discussed until the differences were resolved, changing coding and categories when necessary. 

### 2.5. Trustworthiness

To ensure trustworthiness, that is, credibility, transferability, dependability, and confirmability, several measures were used [26]. Participants were selected from different degrees and ages in the university. Interviews were carried out in a rigorous way letting participants take time to provide their answers and using a conversational style to follow their way of providing the information and adjusting to their rhythm. The topic list elaborated from the literature and reviewed by experts in the field ensured that the main topics regarding UP at this stage in participants’ lives were dealt with. This topic list was not forced onto participants but rather used to stimulate discussion if participants did not address some of the aspects. Finally, interviews were recorded and transcribed for the analysis, and two researchers carried out the analysis. Transferability was guaranteed using a theoretical framework for data analysis to ensure theoretical generalizability, and quotations from participants inform how it was identified in the context of this study. Last but not least, reflexivity was used throughout the whole research process [27]. Regarding reflexivity, the different backgrounds of the authors, combining academic and clinical expertise in the field of health promotion, adolescents, and sexual health, permitted fruitful discussions, but were considered as a threat to collecting participants’ own views in the reflexive process. Meetings were held to prepare data collection and make sure that interviews were carried out with participants feeling comfortable to share their opinions. During the analysis process, reflexivity was used, creating written reflexive notes about interpretations and continuing with discussions between researchers about their understanding of the data.

## 3. Results

The findings are presented grouping the main identified needs into those related with capability, opportunity, and motivation, according to the COM-B Model [13]. This model comprises capability (physical and psychological), opportunity (social and physical), and motivation (automatic and reflective). The authors [13] proposed that people need these three factors to enhance the likelihood of performing a behaviour. In another sense, the model is not only able to address needs but also to design interventions to satisfy them. We decided to base the research on this model because it allows us to both address the needs of the participants, therefore answering the research question, and to design future interventions to satisfy their needs. 

### 3.1. Needs Related to Capability (Physical and Psychological)

Families play a key role in sexual education. The participants’ discourse about their experience in the family environment is heterogenous. On the one hand, there are families who consider this a taboo subject, while other participants said it was fully addressed by their families. For example, one of the participants declared: “I, for example, speak openly with my mother, but I know a lot of people who don’t” (P13). Another participant said: “It’s, you know, a taboo topic. It depends on your family, I can talk about it without a problem, I do, but I know families that, I have friends whose families do not talk about this topic, because it’s taboo …, as it’s related to sex and all that, then it’s taboo” (P3). 

This mainly depends on the type of relationship that the adolescents have with their parents, as sometimes they understand their parents think of them as too young to address this topic. In this sense, one of the participants said that her parents “try to avoid these issues because it’s like ‘no, you’re not going to do that because, you are a child, you are young, I don’t know, you’ll have the time’ but well, in the end, it’s something we don’t [talk about]” (P9).

Sexual education during the school years could be improved. It is addressed in talks which mainly focus on well-known contraceptives by young people, such as the male condom or oral contraceptives, but is not adapted by age and/or maturity. For example, P2 declared: “talks in school are great, but… I think that it should be addressed at home too because I don’t think a talk at school every three years helps too much”. P4 said: “we generally don’t have much information, with just a one hour talk a year and in two secondary school years I think it’s not enough at all”. P6 shared the same opinion: “sometimes, we would get a lesson, well, one of those workshops about sexuality where they talked to us, well, a bit about contraceptive methods and… all that. But, well, we didn’t get much information”.

They think it would be convenient for these talks to be delivered by healthcare professionals, who are, according to the discourse, the most qualified people to address this topic. A participant described it in the following way: “I mean, a nurse, a doctor, or a sexologist, no one will know more about it than them. They are constantly updating, they know. Take it when you’re 15, they know which contraceptive method is going to be better for you… or which one is the riskiest for you, or… how careful you have to be” (P7). Another adolescent argued that “who would be better than a healthcare professional, a nurse or a doctor, to talk about these issues and say ‘look, this happens, we know these methods, and this happens with them’” (P9). 

There is general agreement about the need to incorporate formal sexual education before higher education. For example, P7 said, after being asked about what young people need to prevent UPs, that the first thing would be “a subject since primary school called ’sexual life’ or something sexuality-related […]”.

Lack of information by reliable sources results in female adolescents consulting digital resources (the internet, social networks, or apps) and their peers. Several participants described this in the following way: “There are Instagram profiles of women that like advertise, um, well, promote, mm… relationship and sexuality topics” (P9); “I mean, it’s true that sometimes, I’ve had to really look for information by specialists. There’s a girl on Instagram that specializes in uploading videos and information about… well, sexuality and those things” (P3). 

Even though all participants agreed that the aforementioned sources are not the best or more reliable, these are the most used due to their accessibility, and they allow proactive and specific searches tailored to their needs. Hence, P4 believed that young women use the internet as a source of information “because the topic is then addressed on social media, which is something that almost each of us has access to”, and P12 thought that the most used source is the internet, and also said “It’s quicker, and logically, you can find anything, good information, or a webpage that says something, and then another webpage that says another thing and in then you may end up doing something wrong”.

### 3.2. Needs Related to Opportunity (Social and Physiological)

The participants were aware of the availability of social and health resources (primary care and family planning) that they could use to cover any of their needs about sexual health, such as the use of contraceptives. However, the participants considered that it is necessary to promote a social context change to improve their use; if adolescents normalize the use of the sexual health resources, they could be used without developing feelings or emotions in the adolescents that act as barriers. Some participants described being too embarrassed to talk about their personal issues. In this sense, P6 declared that young people do not use these resources “because of shame, because they’re embarrassed on face-to-face visits, because they’re so young and talk… get information about… about these issues”. P9 also provided a similar response: “because they’re ashamed or something”. P13 said: “for example, I know a girl who didn’t want to go to Family Planning because she was scared to talk to a stranger, and she was scared her parents would eventually find out, and of being judged, for visiting Family Planning”.

The participants suggested they should have access to professionals who would specifically advise them about which contraceptive to use. For example, by bringing the healthcare system closer to their population group with talks in schools, with verbal or written information when visiting Primary Health Care centres for other reasons, or by mailing information to their homes. P7 said “my primary care doctor, and my primary care nurse, none said ‘listen, I’m going to give you this handout about sexual relations’, or gave me a talk, nothing”. P13 added: “if they would do more talks, that would bring the healthcare system closer to people. Maybe they would see it as something approachable and they would not see it as something far away from their reality”. 

The most commonly used contraceptive methods are condoms and oral contraceptives, and occasionally both are combined. The participants cite a correct and consistent use of contraceptives, although they admit knowing other young people who do not use any contraceptive methods, instead using withdrawal (pulling out) or emergency contraception. The participants who were in a stable relationship preferred to always carry condoms and those who were not preferred to buy them when needed.

Both family and friends stand out as sources of social support. Talking to female friends and family is generally a very important source of support to develop behaviours that prevent UPs, as they trust their recommendations and they imitate their behaviours. P2 described “parents… I don’t know, you trust them and so if they say something it’s because of a reason”. P1 stated that “being surrounded by… people with certain behaviours, then you, eventually, are going to imitate those behaviours, aren’t you?”.

The contraceptive methods are expensive for participants. They believe that UP could be prevented if these methods were more affordable. For example, P11, said “If the cost of contraceptive methods was lower…, because you have to buy them if you want to take precautions” and P6 described that “sometimes it [cost] influences because there are people who can’t afford to spend that money”.

### 3.3. Needs Related to Motivation (Reflective and Automatic) 

Partners are an important influence on behaviours for the prevention of UP, especially on the use of contraceptive methods. This influence can be bidirectional. Mutual trust and effective communication generally lead to an agreement on which contraceptive methods to use. For example, P9 declared that her sexual partner plays a key role in the maintenance of healthy sexual behaviours and in the use of contraceptives as “It’s the both of us who talk and make a decision ‘well, then we are going to do this’”. On the contrary, partners can persuade women to not use contraceptive methods. The persuasion usually leads to negative feelings and insecurity in women. P5 said that, in the case of a partner who does not want to use contraceptive methods “maybe he’ll make you give in, and you say ‘jeez, poor thing’ or ‘it’s ok, because if I say I only want to do it using a condom, then he won’t want to do it’ or things like that”. P12 expressed similar views, “you may think ‘well, then if I don’t do it, he will get angry, or if I don’t do it, I don’t know’ and that’s when you give in”. P13 believed that “if one of the partners pushes the issue, then one can give in and say ‘ok, then we won’t use it’”.

Participants believe that society tends to blame women for UPs. Some of them even believe that had they become pregnant during their adolescence, they could have been criticized or rejected by society. In this sense, P6 said “if anything happens or something, the woman [is] blamed then, if you do not use a condom too”. P3 believed that one of the potential consequences of an UP is being judged by society “even though it could have been an accident, you’re going to be judged as [if] you got pregnant for not using a condom. But you’re being criticized by someone who does exactly the same but was lucky not to have it happen to her, you know?”. P10 said that “you’ll get judged a lot for an unwanted pregnancy in your teens”.

## 4. Discussion

The results of the present study have revealed the needs of first-year female university adolescents regarding the prevention of UP. Among these needs we highlight the need for education, for young women and also for their families and partners, by healthcare professionals; accessibility to non-face-to-face healthcare resources and condoms; mutual trust and adequate communication in relationships with their male partners; and a better understanding of pregnant adolescents by society.

This research study offers significant theoretical and practical implications. Considering that unwanted pregnancies continue, in our times, to be an important public health problem that needs to be urgently addressed, the results of this research highlight the needs that young women have in relation to this issue. To this end, this study can guide health professionals and universities to improve the quality of care and services offered, and thus help university students prevent unwanted pregnancies.

The participants highlighted the essential role of families in the sexual education of adolescents. However, they declared that not all families address this topic. This seems consistent with the literature. Padilla-Walker [28] indicates that there is a high percentage of adolescents who do not engage in any form of communication about sexuality with their parents. This goes against the recommendations of several authors, who emphasize the importance of an effective parent–adolescent communication as it results in protection against the development of risky sexual behaviour [29,30]. Among the recommended strategies for effective communication is the involvement of both parents, not just the mother. It is necessary that both parents talk to adolescents even when parents do not consider their offspring at risk [28,30] and to avoid falling into patronizing discourses or reprimands as this could have the opposite effect to the intended one, that is, that adolescents engaging in sexual intercourse [31]. Lack of education and training and lack of trust are among the most relevant reasons why parents do not talk to adolescents about sexuality [32].

This context may result in families believing that schools are the appropriate place for sexual education. However, as shown by the discourse of the participants, sexual education in schools is almost non-existent and it is limited to very few talks. This situation is not surprising because the educational laws that have existed in Spain did not include anything regarding education on health issues. The new education law approved in 2020, LOMLOE, for the first time specifically includes the obligation to address sexual health. Textually it indicates “*In the same way, education for consumption will be worked on responsible and sustainable development, health education, including sexual health education*” [33]. This situation ignored the recommendations of the World Health Organization [34], promulgated in 2002, which highlighted the importance of implementing educational programs at an early age.

The lack of information is surprising when evidence shows that multicomponent interventions in schools combining school policy changes and parent involvement, among others, are effective for the promotion of sexual health among youths [35], delaying sexual debut and preventing UPs [36]. Perhaps a closer work of public health professionals with primary health care teams will facilitate the implementation of programs that are relevant to adolescent university students. The participants in this research identified healthcare professionals as the most adequate sexual educators during the school years. Other studies have already identified this preference by adolescents and parents. More specifically, they identified primary care professionals as the preferred source of health information during adolescence, being sexuality one of the topics covered [37]. Other studies analyse more deeply the role healthcare professionals as potential sexual educators, as they can develop brief interventions to promote healthy sexual behaviours among adolescents, such as the use of condoms [38], the delay [39] or intention to delay sexual debut [26], or effective parent–adolescent communication [38,40] that can be implemented by these care services. Furthermore, healthcare professionals could support families by addressing lack of knowledge or lack of trust when communicating with adolescents about sexuality [32]. This could have a positive effect in other areas of adolescent health by addressing parent–adolescent communication during this life stage.

In the absence of information or communication from their parents and formal sexual education, adolescents rely on other sources of information, such as the internet or mobile apps. Such a reality is already covered by the literature [40] and it is not surprising taking into consideration the widely accepted use of mobile phone technologies among adolescents [41]. In fact, there are specific mobile apps about sexual health [42]. With the proliferation of information in this format, it is extremely important to establish systems for the identification of digital resources that can be recommended, and therefore used, by adolescents.

According to COM-B model, educational interventions, and therefore knowledge acquisition, contribute to improve psychological capability as well as reflective motivation. Both are directly related with the development of healthy behaviours [13]. 

The participants know about healthcare resources they can freely use when needed. However, their emotional state, such as feelings of shame or fear, which prevents them from using the services stands out. Cassidy et al. [43] already described these motivations, with stigmas, shame, or discomfort acting as barriers that prevent the use of these healthcare services. Some authors have developed interventions focused on eliminating these barriers and encouraging the use of these healthcare resources [16]. They recommend personalized, multilevel interventions with a facilitative approach based on the individual needs of each person [15]. 

The risk of UP is related to different behaviours, with the non-use of condoms standing out [11]. Hence, some authors suggest implementing interventions to specifically promote the use of this contraceptive method [44], as it has already been established that the use of contraceptive methods is essential for sexual health [45]. However, the participants in the present study have identified the high cost of condoms as a barrier. Therefore, it seems appropriate that interventions also include improving access to condoms. In this sense, universities play a key role as they can potentially implement health promotion strategies to prevent UPs among the university population that include both sexual education and access to condoms as well as screening methods [10]. Providing services and resources in an accessible way is another of the interventions included in the theoretical model used in this study as a promoter of the development of healthy behaviours [13].

The role of male partners in the development of risk behaviours that may result in an UP is a cornerstone in female adolescents’ narratives. The positive or negative influence depends on the previous existence of an agreement to use contraceptive methods, as well as on effective communication and mutual trust between both partners. The male partner can play a persuasive role in the promotion of and engagement in risk behaviours, such as sex without condoms. The literature supports this finding and shows that effective communication is essential for an agreement on contraceptive use [46]. More specifically, the non-use of condoms by male partners has been linked to poor knowledge about sexual health and lack of communication [47]. It is probable that the reality perceived by the participants, which seems to replicate the literature, only highlights the significant need for measures to improve sexual education that include both men and women. While the interventions in the reviewed literature do not always include both partners, it is obvious that women are not exclusively responsible for sexual relationships that may result in UPs. This calls for the implementation of measures to promote sexual health that involve both men and women.

## 5. Limitations, Strengths and Future Implications

This qualitative study attempted to explore university students’ understanding and behaviours regarding unwanted pregnancies and identify their needs to prevent them. However, it had several limitations. First, since all participants were university students, it is not possible to make generalizations to other adolescents with different education levels. In addition, the convenience sample obtained may not have been able to encompass all the perceptions of the population regarding the subject under study. Thus, future research should include other samples to obtain additional evidence.

Despite this, this study had its strengths in that it was the first study to be carried out with these characteristics in Spain, providing a more concrete, albeit limited, understanding of behaviours regarding unwanted pregnancies in university students and identifying their needs to prevent them.

## 6. Conclusions

The findings from this qualitative research show the existence of needs related to the prevention of UPs among university adolescents. They are also useful to identify areas and methods for successful interventions that using quantitative methods could not have permitted [16]. The feelings women have, i.e., guilt, shame, and fear, need to be considered when including other meaningful components such as sexual education for female adolescents and their partners, the promotion of sexual education for families, access to non-face-to-face healthcare resources, and access to less expensive contraceptive methods.

## Figures and Tables

**Table 1 healthcare-12-00425-t001:** Age of participants and studies they were undertaking (2022).

Participant ID Number	Age	Studies They Are Doing in the University
P1	19	Early Childhood Education
P2	19	Social Education
P3	19	Psychology
P4	19	Philology
P5	19	Nursing
P6	20	Elementary Education
P7	20	Early Childhood Education
P8	19	Finance
P9	20	Law
P10	18	Translation and Interpretation
P11	19	Spanish Studies
P12	19	Law
P13	18	Social Work

**Table 2 healthcare-12-00425-t002:** Interviews initial topic list * (2022).

What does sexual health mean to you? How would you describe your sexual health? What do you think young women of your age need to prevent unwanted pregnancies?
**SPECIFIC QUESTIONS**
**Questions related to capability**
- What do you think you can do to prevent unwanted pregnancies? And your female friends? - What do you think about the information young women currently have about sexuality/sexual and reproductive health? - What kind of information should be made available to young women your age in order to prevent unwanted pregnancies? - Besides information, what else would you like to be done to help you have a healthy sexual and reproductive life? And for other women?
**Questions related to opportunity**
- Do you currently use any contraceptive method when you engage in sexual activity? Which one(s)? How do you get them? - And other people around you? - What role does your economy play in your access to contraceptive methods? - What role do your friends and peers play in your decision to engage in healthy sexual behaviours and the use of contraceptives?/How are you influenced by what your friends do? - What role does your sexual partner play in maintaining healthy sexual behaviours and in the use of contraceptives?/How are you influenced by what your partner thinks or wants to do? - How do you think society helps young people to prevent unwanted pregnancies? - And what does the healthcare system do about it? - How do you think the environment (family, friends, or health systems) influence the healthy behaviours of young women to prevent unwanted pregnancies?
**Questions related to motivation**
- What would the consequences of an unwanted pregnancy be for you? - How do said consequences influence your decision to adopt behaviours in order to prevent unwanted pregnancies?
Anything else you would like to add?

* The topic list was built using the previous literature and suggestions of experts and members of the Regional and National Campaign to Prevent Unwanted Pregnancy.

## Data Availability

The datasets generated and/or analysed during the current study are not publicly available due to data privacy but are available from the corresponding author upon reasonable request.

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
