# Peer review of "Young Women’s Needs Regarding Sexual Preventive Behaviours and Unwanted Pregnancies"

_healthcare, 2024, doi:10.3390/healthcare12040425_

Round 1

Reviewer 1 Report

Comments and Suggestions for Authors

Dear authors,

It was a great pleasure to have the opportunity to read and review your work. It is very interesting research, on a relevant topic to the area and with a design that could support other investigations.

To improve the work, I make the following suggestions and comments:

Abstract: the expression "when so much progress" could be considered a judgement, so I suggest something more neutral.

Introduction: needs development. You could address more the importance of the topic, mains research in the area, the selected model (in favor of others that are also used in this area), and the methodological design used.

Method: indicate the sample selection criteria. The instrument may be characterized with examples in this section and in full in the annex.

The development of the script should be included in the description of the procedures.

Line 118, pg. 4 - change the sentence as it refers to value judgment.

Identify all participants in the speeches (e.g., line 121, page 4 does not have this indication). Line 145, page 4 - in the case of the answer to the same question, the paragraph could be removed.

Discussion: you could develop a discussion of the answers outlined in the model and the impact of using technologies with support in the area. Indicate the limitations of the study.

Conclusions: they could develop further, identifying the potential of work for the different sectors.

Author Response

Abstract: the expression "when so much progress" could be considered a judgement, so I suggest something more neutral.

Introduction: needs development. You could address more the importance of the topic, mains research in the area, the selected model (in favor of others that are also used in this area), and the methodological design used.

 Thanks for both suggestions. We have rewritten the abstract, and moreover have checked over and made major edits in the introduction accordingly.

 Method: indicate the sample selection criteria. The instrument may be characterized with examples in this section and in full in the annex.

The development of the script should be included in the description of the procedures.

Line 118, pg. 4 - change the sentence as it refers to value judgment.

Thanks.  We include the instrument and examples in the text. According with previous papers with similar designs, we are going to be able to share the data on request. We are going to declare it in the “Data Availability Statement: Data Availability Statement: The datasets generated and/or analyzed during the current study are not publicly available due to data privacy but are available from the corresponding author upon reasonable request.”  In addition, we have rewritten the text in the line 118.

Identify all participants in the speeches (e.g., line 121, page 4 does not have this indication). Line 145, page 4 - in the case of the answer to the same question, the paragraph could be removed.

Thanks, all speeches have identified the participants, some at the start of the sentence (i.e. P6 line 184 or at the end of the sentence, i.e. P3 in line 171). In any case, we have rewritten the line 121 according with the reviewer suggestion.

Discussion: you could develop a discussion of the answers outlined in the model and the impact of using technologies with support in the area. Indicate the limitations of the study.

Thanks, we have rewritten the discussion, not only where the reviewer suggest, to highlight the link between both, the findings and the theorical model.

Conclusions: they could develop further, identifying the potential of work for the different sectors.

Thanks for the suggestion. We include, according to the recommendations of all reviewers, a new section “Limitations, strengths and future implications” before conclusions.

Reviewer 2 Report

Comments and Suggestions for Authors

Dear Authors,

Congratulations on completing your research and preparing the manuscript for submission. To further raise its standard, I respectfully suggest that you consider the following recommendations:

Title: For greater clarity, consider replacing "views" with "needs."

Introduction.

- It would be beneficial to add information about "needs assessment." Incorporating this type of assessment into the activities of the manuscript, including the integration of individual comments, could enrich your research.

- The introduction could be strengthened by including either the research question (to clarify the purpose of conducting the research), the research assumptions (hypothesis), or, given the qualitative nature of your study, a clear specification of its purpose.

- Adding contextual information, especially about the nature and extent of sex education in schools, would be valuable. As your international audience may not have accurate information about this, such context is crucial to understanding the significance of your findings.

Materials and Methods

- More detailed information about the research design would enhance this section.

- Regarding line 58, please clarify what the "descriptive design" specifically entails.

- On line 61, a more detailed explanation of the mixed methods used would be beneficial.

- For line 66, could you specify the exact method of data collection used?

- Given the sensitive nature of your research, it is critical to include information about interviewers and how respondent characteristics influenced interviewer selection. This has a significant impact on the quality and accuracy of the responses.

- How does the content focus of the interview, as presented in Table 2, align with BCW theory? Could you clarify which specific questions identify each component of this theory?

In section 2.4, it would be helpful to indicate what software was used for data analysis. Since some sources suggest the use of grounded theory (see reference 15), a description of its use in your research is warranted.

Section 3.1 (from line 103)

- Defining "capability" and explaining how it was operationalized in your study would clarify this section.

Section 3.2 (from line 157)

- A clear definition of "opportunity" as used in your research is needed.

- The content in lines 158-176 seems more aligned with "capability" than "opportunity". Could you please revise this?

Section 3.3 (from line 194)

- A definition of "motivation" as it relates to your study would be beneficial.

Discussion

- Integrating the findings with specific practices in schools in your country within the discussion would provide valuable context. Distinguishing between findings that reflect specific situations and those that indicate systemic issues would enrich the analysis.

- With regard to lines 225-231, please explain the novel findings of your research. While some findings may predate the study, it is important to clarify how your research advances knowledge in this area. What new perspectives do your transcripts reveal?

- Adding details about your respondents' social backgrounds may explain certain findings and suggest applicability to other contexts. This is particularly relevant to the discussion of lines 290-292.

- The influence of social desirability bias, mentioned in lines 303-306, should ideally be addressed in the methods section. Its consideration at the research design stage and in the specification of interview procedures is paramount. Introducing it at the end of the discussion seems somewhat misplaced.

- Finally, outlining the limitations of your research and any restrictions on its replicability would provide a balanced view of the scope and applicability of your study.

Sincerely,

Author Response

Title: For greater clarity, consider replacing "views" with "needs."

Thanks, we have rewritten the title.

Introduction.

It would be beneficial to add information about "needs assessment." Incorporating this type of assessment into the activities of the manuscript, including the integration of individual comments, could enrich your research. The introduction could be strengthened by including either the research question (to clarify the purpose of conducting the research), the research assumptions (hypothesis), or, given the qualitative nature of your study, a clear specification of its purpose. Adding contextual information, especially about the nature and extent of sex education in schools, would be valuable. As your international audience may not have accurate information about this, such context is crucial to understanding the significance of your findings.

Thanks for all suggestions. Without doubt they are going to contribute to improve the quality of the introduction. According with the comments of the 3 reviewers we have rewritten the introduction.  All changes appear highlighted in the text.

 Materials and Methods

More detailed information about the research design would enhance this section. Regarding line 58, please clarify what the "descriptive design" specifically entails.

Thanks for the recommendations. We add in red information to answer the previews two comments.  A qualitative descriptive design was used to gain a comprehensive understanding of the phenomenon of UP among female university students. The qualitative descriptive design is suitable for studies that need to provide descriptions of experiences and perceptions in the area under investigation but do not require a deeply theoretical context [12]. This type of design permits to access to participants views as they present them, recognizing the subjective nature of questions under research (Bradshaw et al., 2017) [19]. This is particularly relevant in healthcare research, which is commonly concerned with how participants experience health and illness and the associated healthcare interventions [18]. This approach is recommended when it is necessary to access the subjective nature of the problem, with a primary focus on enhancing practice or obtaining data for use in mixed methods studies to design and evaluate interventions [18]. Therefore, this type of design permits not only to access to participants views but to obtain data to enhance sexual health promotion practices and develop interventions as well.

On line 61, a more detailed explanation of the mixed methods used would be beneficial.

Thanks for the suggestion. The study reported here is not a mix method study. This information is provided to explain when the descriptive approach in suitable for.  

For line 66, could you specify the exact method of data collection used?

Thanks. We have rewritten the text according with you recommendation. Semi-structured interviews (Kallio et al. [2016] were carried out using a conversational style [21]. Namely, an initial prompt was presented, and the answers provided were further explored using a topic guide when the participants’ answers were short or vague (Table 2). The questions included aspects identified in the literature to be relevant to address UP. The list of potential questions was reviewed by research team members to ensure clarity, relevance and completeness. This list was used as the preliminary guide, and the possibility of reformulating the questions as the interviews took place was considered by the research team. Participants responded without difficulties and not changes were necessary. The interviews were digitally recorded and transcribed verbatim.

Given the sensitive nature of your research, it is critical to include information about interviewers and how respondent characteristics influenced interviewer selection. This has a significant impact on the quality and accuracy of the responses.

Thanks. In effect, a sensitive topic was addressed in the research. So, we were extremely cautious in all methodological aspects to avoid causing discomfort to the participants and biases. We include the following text in the page 4 according to your suggestion “Due the sensitive nature of the topic addressed, and, in order to empathize with the participants and to avoid that the answers could be conditioned by feelings, such as embarrassment, the interviews were conducted by a 32-year-old woman, a nurse specialized in family and community care, with a high level of experience in the subject, not only from a scientific perspective but also from a real-life perspective as she works in a paediatric and adolescent primary care unit”

How does the content focus of the interview, as presented in Table 2, align with BCW theory? Could you clarify which specific questions identify each component of this theory?

Thanks, we include the components of the model corresponding to the questions.

In section 2.4, it would be helpful to indicate what software was used for data analysis. Since some sources suggest the use of grounded theory (see reference 15), a description of its use in your research is warranted.

We did not use any software for data analysis. The identification of codes categories and themes was done in printed transcriptions of the interviews and using Word (the review tools such as comments and other tools such as highlighting and format). That was the reason not to report the software.

The only reference related to grounded theory was used to refer to saturation in data analysis. It was not used a grounded theory approach.

Section 3.1 (from line 103) Section 3.2 (from line 157) Section 3.3 (from line 194). Defining "capability" and explaining how it was operationalized in your study would clarify this section; A clear definition of "opportunity" as used in your research is needed; A definition of "motivation" as it relates to your study would be beneficial.

Thanks for the suggestion. We include a common paragraph in at the beginning of the section 3 to answer all suggestions.

The content in lines 158-176 seems more aligned with "capability" than "opportunity". Could you please revise this? 

Thanks. In effect it could be interpreted as capability, but the discourse of the participants was more aligned with changing the social context. It was clearly interpreted by us. Nonetheless, to clarify this point we have rewritten the section 3.2.

Discussion

Integrating the findings with specific practices in schools in your country within the discussion would provide valuable context. Distinguishing between findings that reflect specific situations and those that indicate systemic issues would enrich the analysis; With regard to lines 225-231, please explain the novel findings of your research. While some findings may predate the study, it is important to clarify how your research advances knowledge in this area. What new perspectives do your transcripts reveal?; Adding details about your respondents' social backgrounds may explain certain findings and suggest applicability to other contexts. This is particularly relevant to the discussion of lines 290-292; The influence of social desirability bias, mentioned in lines 303-306, should ideally be addressed in the methods section. Its consideration at the research design stage and in the specification of interview procedures is paramount. Introducing it at the end of the discussion seems somewhat misplaced; Finally, outlining the limitations of your research and any restrictions on its replicability would provide a balanced view of the scope and applicability of your study.

Thanks for all suggestions. We have modified the discussion section according with the adequate recommendations of the reviewers.

Reviewer 3 Report

Comments and Suggestions for Authors

Dear Editor,

Thank you very much for the opportunity to review the manuscript. Below are the comments for this manuscript:

Abstract: Please write the themes or categories that you found as the findings of the study.

Introduction:

The introduction is not adequate. For example, in Paragraph 2, line 42, “Most of the interventions so far include…” Interventions about what? Intervention to reduce unwanted pregnancy, or how? Please elaborate more. Writing the sufficient information when citing studies will help readers to get more comprehensive understanding of what authors meant to convey.

The gap of knowledge is also not clearly described. Why did the authors conduct this qualitative descriptive study? What is the significance of this study?

Materials and Methods:

What are the reasons for deciding the inclusion criteria of not being previously or currently pregnant?

This manuscript needs a section for data collection. Please explain how authors collected the data: How did researchers recruit the participants? Where did the interviews take place? In which country was the study conducted? How were interviews conducted, e.g., face-to-face interview, or online, video call, phone, etc.? Who interviewed the participants? How long did it take for one interview? When were the data collected? In what language was the interview performed? If in the language other than English and was reported in English, how did the authors translate the data (excerpts, themes, or categories) reported in this manuscript? 

Page 4, line 90, it is written “A framework analysis approach was used for data analysis.” Please elaborate more on what framework analysis approach was used and according to whom. Please provide the citation.

How did the authors maintain the trustworthiness of the study? Please make a section about trustworthiness of the study. 

Results: Authors presented three main findings. Are these themes, or categories, or groups, or how did the authors name these three findings? Please explain for clarity.

Discussion: As the authors mentioned on page 2, line 54, that BSW theory is used in this study, please discuss the link between the findings of the study and the BSW theory.

Please add the strengths and limitations of the study, the implications of the study, and the recommendations from this study.

Author Response

Abstract: Please write the themes or categories that you found as the findings of the study.

Introduction: The introduction is not adequate. For example, in Paragraph 2, line 42, “Most of the interventions so far include…” Interventions about what? Intervention to reduce unwanted pregnancy, or how? Please elaborate more. Writing the sufficient information when citing studies will help readers to get more comprehensive understanding of what authors meant to convey; The gap of knowledge is also not clearly described. Why did the authors conduct this qualitative descriptive study? What is the significance of this study?

Thank for the suggestion. We have rewritten both, the abstract and introduction, according with your (and other reviewers) recommendations. 

Materials and Methods:

What are the reasons for deciding the inclusion criteria of not being previously or currently pregnant?

Thanks, the reason was that the perception of the needs could be mediated by a previous pregnancy, and it could have a negative and significant impact on the quality and accuracy of the responses.

This manuscript needs a section for data collection. Please explain how authors collected the data: How did researchers recruit the participants? Where did the interviews take place? In which country was the study conducted? How were interviews conducted, e.g., face-to-face interview, or online, video call, phone, etc.? Who interviewed the participants? How long did it take for one interview? When were the data collected? In what language was the interview performed? If in the language other than English and was reported in English, how did the authors translate the data (excerpts, themes, or categories) reported in this manuscript? 

Thanks for such a constructive comment. We have rewritten the material and method section according to its comments.

Since the research was developed in Spain and in Spanish language, the data analysis was done also in Spanish. The translation into English to include the information in the paper was developed by a professional translator.

Page 4, line 90, it is written “A framework analysis approach was used for data analysis.” Please elaborate more on what framework analysis approach was used and according to whom. Please provide the citation.

Apologize for the lack of information, we have added the citation “A framework analysis approach was used for data analysis following the phases of familiarization with the data and the theoretical framework, the codification of the transcribed interviews, and the interpretation and grouping of the codes into categories (Ritchie & Spencer [25] To get familiar with the data, every interview was read at least 3 times. The first reading helped to identify words that were related to the understanding of UP and the participants’ prevention behaviors. In a second reading, codes were created to describe the content of each fragment of the interviews. The codes were afterwards grouped into categories and themes according to the concepts of the COM-B Model [8] with flexibility to identify possible new categories, if the data required it, although no new categories were needed. Two researchers analyzed separately the data, and disagreements were dis-cussed until the differences were resolved, changing coding and categories when necessary”.

How did the authors maintain the trustworthiness of the study? Please make a section about trustworthiness of the study. 

Thanks for the suggestion. Section 2.5 Trustworthiness was included.

Results: Authors presented three main findings. Are these themes, or categories, or groups, or how did the authors name these three findings? Please explain for clarity.

Thanks, we clarify this point at the beginning of the section 3.

Discussion: As the authors mentioned on page 2, line 54, that BSW theory is used in this study, please discuss the link between the findings of the study and the BSW theory.

Thanks, we have rewritten the discussion, not only where the reviewer suggest, to highlight the link between both, the findings and the theorical model.

Please add the strengths and limitations of the study, the implications of the study, and the recommendations from this study.

Thanks for the suggestion. We include, according to the recommendations of all reviewers, a new section “Limitations, strengths and future implications” before conclusions.

Round 2

Reviewer 2 Report

Comments and Suggestions for Authors

Dear Authors,

Having carefully reviewed your latest submission, I am delighted to observe that you have adeptly and fully addressed each of the comments and concerns previously highlighted. Your meticulous efforts and commitment to enhancing the manuscript are truly praiseworthy.

At this juncture, I have no additional comments to add. Observing the progression of your work has been a rewarding experience. I extend my sincerest best wishes for your continued success and am confident that your research will contribute significantly to our field.

Warm regards,

Author Response

Dear reviewer,

We appreciate your valuable review. Since your comments do not suggest any additional modifications we are simply writing to thank you for your time and effort.

Kind regards,

Reviewer 3 Report

Comments and Suggestions for Authors

Dear Editor,

Thank you very much for the opportunity to review the revised manuscript. The authors made substantial revisions to address our previous comments. However, there are some minor comments that have not been addressed, especially in the method part. In the previous comments, we suggested authors to add a section for data collection process and explain how the data were collected, for example: how authors approached the participants, when the interviews took place (from which month which year to which month which year), how long each interview took place (for example 30 minutes, one hour, etc). We suggest authors too change "measures" section to "data collection," because this is a qualitative study, and add the missing information that have been mentioned above. 

In the last sentence of new section "trustworthiness," it is stated that "reflectivity was used along the whole research process." This statement needs more information. How did authors perform/use reflectivity in the whole process?

Thank you very much and good luck for the revision.

Author Response

Dear reviewer. We appreciate your valuable review. Below we respond to your suggestions and have included the changes in the manuscript.

 In the previous comments, we suggested authors to add a section for data collection process and explain how the data were collected, for example: how authors approached the participants, when the interviews took place (from which month which year to which month which year), how long each interview took place (for example 30 minutes, one hour, etc). We suggest authors too change "measures" section to "data collection," because this is a qualitative study, and add the missing information that have been mentioned above.

Thanks for the recommendation. We have rewritten the name of the section.

In the last sentence of new section "trustworthiness," it is stated that "reflectivity was used along the whole research process." This statement needs more information. How did authors perform/use reflectivity in the whole process?

Thanks for the comment. We have included more information.